# Effect of Peptides from Plasma of Patients with Coronary Artery Disease on the Vascular Endothelial Cells

**DOI:** 10.3390/medicina59020238

**Published:** 2023-01-27

**Authors:** Marko Kozyk, Kateryna Strubchevska, Tetiana Marynenko, Alena Zlatska, Tetiana Halenova, Nataliia Raksha, Olexii Savchuk, Tetyana Falalyeyeva, Oleksandr Kovalchuk, Ludmyla Ostapchenko

**Affiliations:** 1Corewell Health William Beaumont University Hospital, Royal Oak, MI 48073, USA; 2Educational and Scientific Centre “Institute of Biology and Medicine”, Taras Shevchenko National University of Kyiv, 64/13 Volodymyrska Str., 01601 Kyiv, Ukraine; 3State Institute of Genetic and Regenerative Medicine NAMS of Ukraine, 67 Vyshgorodska Str., 04114 Kyiv, Ukraine

**Keywords:** coronary artery disease, peptide pool, endothelial cells, endothelin-1, von Willebrand factor, tissue plasminogen activator, plasminogen activator inhibitor-1

## Abstract

*Background and Objectives*: Coronary artery disease (CAD) is the foremost cause of adult disability and mortality. There is an urgent need to focus on the research of new approaches for the prevention and treatment of CAD. *Materials and Methods*: The effects of peptides isolated from the blood plasma of CAD patients on endothelial cell secretion using the in vitro model have been tested. Human endothelial progenitor cells (HEPCs) were incubated for 24 h with peptides isolated from the plasma of healthy subjects or patients with stable angina, progressive unstable angina, and myocardial infarction. The contents of some soluble anticoagulant as well as procoagulant mediators in HEPC culture treated with peptide pools were then compared. *Results and Conclusion*: The results show that peptides from the plasma of patients with myocardial infarction promote endothelial cells to release both von Willebrand factor and endothelin-1, increasing vasoconstriction and shifting hemostatic balance toward a prothrombotic state. In contrast, peptides from the plasma of patients with progressive unstable angina suppress the secretion of endothelin-1 by HEPCs, while the secretion of both von Willebrand factor and tissue plasminogen activator was increased. As can be seen from the results obtained, disease derived peptides may contribute to the homeostasis of living organisms or the progression of pathological processes.

## 1. Introduction

Coronary artery disease (CAD) is the most common type of heart disease all over the world [1]. CAD is caused by plaque buildup in the inner lining of coronary arteries that supply blood to the heart. Plaque is made up of cholesterol deposits and other substances including cellular waste products, calcium, and fibrin. As the plaque builds up in the process known as atherosclerosis, the artery wall becomes thickened and the artery lumen narrows over time, which can partially or totally block the blood flow [1,2]. To date, there are numerous risk factors that have been identified to be responsible for causing atherosclerosis. These factors include dyslipidemia, diabetes, obesity, smoking, family history, sedentary lifestyle, etc., [3,4]. Despite a significant focus of many researchers on atherosclerosis, the mechanisms of the formation of atherosclerotic lesions are still not fully understood. However, numerous studies have shown that coronary endothelial dysfunction contributes to both the initiation and progression of atherosclerotic plaques [5,6].

Over the last two decades, it has been shown that vascular endothelium is an active autocrine, paracrine, and endocrine organ that lines the entire circulatory system from the heart to the smallest capillaries, and plays an essential role in the regulation of vascular tone and the maintenance of vascular homeostasis [7]. Under normal conditions, the endothelium not only provides a highly selective physical barrier to control the vascular permeability, but it also releases a large number of vasoprotective and thromboresistant molecules, which maintain the state of vasodilatation over vasoconstriction [7,8]. In contrast, during vascular injury, the release of endothelium-derived constricting and prothrombotic factors has a pivotal role in all phases of thrombus formation. The constitutive synthesis and secretion of vasoactive substances by endothelial cells appear to be under tight regulation as their levels are fairly constant in healthy subjects [8,9]. However, due to mechanisms which are not yet fully understood, the alteration of the normal homeostatic properties of the endothelium may occur and lead to unfavorable physiological vascular changes. Several studies have shown that alterations in endothelial properties can be induced by some endogenous substances such as cytokines, modified proteins, and antibodies, indicating the potential relevance of these mechanisms to cardiovascular events [2,10]. The research of molecules that may affect endothelial function promises a proper approach to find new biomarkers for the prediction of cardiovascular events.

Endothelial cells are constantly exposed to the influence of substances present in blood. One hypothesis is that peptides, among other molecules present in the circulation, are more likely to affect the endothelial function. It is known that the amount and repertoire of peptides in biological fluids change dynamically according to the physiological or pathological state of an individual [11,12]. The appearance of specific plasma peptides in CAD may induce the release of vasoactive molecules by endothelial cells, which are able to shift homeostatic balance towards the pro-coagulant phenotype. On the other hand, as a part of compensatory response under thrombogenic condition, such peptides may affect endothelium to release pro-fibrinolytic molecules to maintain a hemostatic balance. Thus, the aim of this study was to characterize the effects of peptides collected from the blood plasma of healthy subjects and patients with angina or myocardial infarction on endothelial cell secretion using the in vitro model.

## 2. Materials and Methods

### 2.1. Study Groups

The study included 70 patients who were hospitalized to the cardiology department of Kyiv City Hospital #12 with a preliminary diagnosis of CAD. To establish the diagnosis, the guidelines of European Society of Cardiology 2020 were followed [13]. A set of routine laboratory tests, including lipid profile, and total protein, was performed by the laboratory of the hospital. Cardiac troponin I (cTnI) was analyzed by immunoassay (MyBioSource Inc., San Diego, CA, USA; the upper reference limit (99th percentile) was determined at 14 pg/mL, reference ranges: <7 pg/mL for males, and <5 pg/mL—for females). Body mass index (BMI) was calculated as weight in kilograms divided by height in meters squared. After the full set investigation was completed, we defined three groups: patients with stable angina (SA, *n* = 25), patients with progressive unstable angina (PUA, *n* = 28), patients with myocardial infarction (MI, *n* = 17). The control group included 20 healthy volunteers, who did not have cardiovascular or any chronic disease, and were well matched with age, gender, and other basic parameters with experimental patients’ groups. The Ethical Committees of both the Kyiv City Hospital #12 (Kyiv, Ukraine) and Taras Shevchenko University of Kyiv (Kyiv, Ukraine) approved the study and all patients gave informed consent to participate according to the Declaration of Helsinki.

### 2.2. Blood Sampling and Plasma Preparation

Blood samples were drawn from the cubital vein on the day of admission, prior to beginning treatment. Blood samples were collected in standard plastic vacuum tubes with 3.8% sodium citrate. The volume ratio of blood to citrate was equal to 9:1. Plasma was separated by centrifugation at 2500× *g* for 25 min. After centrifugation, all plasma samples were aliquoted, and frozen at −80 °C until used.

### 2.3. Plasma Peptide Pool Isolation

The peptide pool was obtained from plasma according to the procedure described by Nikolaichyk et al., as cited in Katrii et al. [14]. Cold 1.2 M HClO_4_ was added to the 1 mL plasma sample (1:1, *v*/*v*), the mixture was placed on ice for 15 min and centrifuged at 10,000× *g* for 20 min 4 °C. The protein precipitate was discarded. The resultant supernatant was neutralized by 5 N KOH to pH 7.0 and was kept on ice for 15 min, followed by centrifugation. The remaining protein in the resultant supernatant was precipitated with four volumes of cold 96% ethanol and was kept at 4 °C for 30 min. Then, the sample was subjected to centrifugation step again. The supernatant was obtained and its optical density (OD) was measured at 210 nm using the spectrophotometer (Smart SpecTMPlus, BioRad, Hercules, CA, USA). The peptide concentration was calculated using calibration curve prepared with CBZ-glycylglycine dipeptide (0.26 kDa) as a standard.

The purity of peptide fraction was estimated by 15% polyacrylamide gel electrophoresis [15]. Gels were stained with 0.125% solution of Coomassie Brilliant Blue G-250 (Thermo Fisher Scientific, Waltham, MA, USA) in 25% isopropanol and 10% acetic acid.

Peptide fractions separated from 1 mL of the blood plasma of all patients and healthy donors were lyophilized (LyoQuest, Terrassa, Spain) and were kept at 4 °C. Before assay, dry peptide material was dissolved in 0.2 mL of 0.05 M Tris-HCl, pH 7.4, containing 0.13 M NaCl. 

The further analysis of effects of peptides on endothelial cell secretion was repeated at least three times for each experimental group. Every time we used freshly prepared peptide pool solution obtained by mixing of five different peptide fractions isolated from individuals of the same experimental group.

### 2.4. Human Endothelial Progenitor Cell (HEPC) Culture

All experiments with cell culture were performed in accordance with the bioethics and biological safety norms confirmed by the permission of medical company “Good cells” (Kyiv, Ukraine).

Mononuclear cells (MNCs) from human peripheral blood samples were isolated [16]. 20 mL of heparinized venous blood was centrifuged in Histopaque^®^-1077 density gradient (Sigma, St. Louis, MO, USA) at 400× *g* for 30 min at 4 °C. MNCs were washed twice in phosphate-buffered saline (PBS), pH 7.4. The obtained MNCs were seeded onto collagen-coated 75 cm^2^ culture flasks (SPL, Pocheon-si, Korea) in endothelial growing medium with the following composition: MCDB 131 medium (Gibco, Loughborough, UK), 5% fetal bovine serum (Sigma), 2 U/mL heparin sodium (Indar, Kyiv, Ukraine), 1 ng/mL vascular endothelial growth factor, 10 ng/mL epidermal growth factor, 2 ng/mL basic fibroblastic growth factor, 20 ng/mL insulin-like growth factor; 0.2 μg/mL hydrocortisone, and 1 μg/mL ascorbate-2-phosphate (all manufactured by Sigma). Cells were subcultured with a 0.1/0.02% trypsin/ethylenediaminetetraacetic acid mixture in PBS (Sigma).

Studies were performed on secondary cultures grown in 24-well cell culture plates (Becton, Dickinson and Company, Franklin Lakes, NJ, USA) at a density of 3 × 10^3^ cells/cm^2^ under the same conditions as primary cultures. The lyophilized peptide pools from all studied groups were diluted with culture medium. Cells were treated with peptides for 24 h. The final concentration of peptides in culture medium was the same for all experiments (300 µg/mL). The aliquots of culture medium were collected after the appropriate incubation period with peptides (1 h and 24 h), centrifuged at 15,000× *g* to remove cell debris, and frozen at −70 °C until use for immunoassay. The samples of control culture medium that was not exposed to any treatment were collected at the same time periods to analyze the basal level of endothelial cells secretion. The integrity of endothelial cells after incubation with peptide fractions was verified using a commercial kit for measurement of lactate dehydrogenase (LDH) activity (Felicit Diagnostics, Dnipro, Ukraine).

### 2.5. Enzyme-Linked Immunosorbent Assay (ELISA)

Samples of HEPC culture medium in volume of 100 µL were incubated in wells of 96-well plate overnight at 4 °C until analysis the next day. The coated plate was washed three times with wash buffer—PBS, pH 7.4 that contained 0.05% Tween-20. Then, the microtiter plate was incubated with a blocking buffer—PBS, pH 7.4, containing 5% non-fat milk at 37 °C for 60 min to inactivate unspecific binding sites. Monoclonal mouse antibodies, namely, anti-tissue plasminogen activator, anti-plasminogen activator inhibitor-1, anti-von Willebrand factor, and anti-endothelin-1 (all manufactured by Santa Cruz Biotechnology, Dallas, TX, USA) at a dilution of 1:3000 was added to determine corresponding antigens. Anti-mouse secondary antibody conjugated to horseradish peroxidase (Bio-Rad) was used at a dilution of 1:7000. Substrate development was performed with chromogenic mixture—*o*-phenylenediamine (Sigma) and H_2_O_2_ in 0.1 M sodium citrate buffer, pH 5.0. The reaction was terminated by the addition of 1 N H_2_SO_4_. Photometrical evaluation took place with the help of a computer-controlled microplate reader (μQuant^TM^, BioTek Instruments, Inc., Winooski, VT, USA) at a wavelength of 492 nm.

The levels of studied molecules in culture medium that was not exposed to peptide treatment (the volume of 0.05 M Tris-HCl, pH 7.4, containing 0.13 M NaCl, was added instead of peptide solution) at a time point of 1 h were used as a standard, defined as 100% value, for each substance in our experiments. The data obtained at 24 hours’ time point from untreated culture as well as the data for all peptide-treated cultures at 1 h and 24 h were compared to the data from untreated culture at 1 h (Appendix A).

### 2.6. Statistical Analysis

Statistical analysis was performed with Statistica 8.0 software. All experiments were performed by utilizing parallel design and repeated at a minimum of three times each. The data from experiments were expressed as mean ± standard deviation (SD). Data distribution was analyzed using the Kolmogorov-Smirnov test. Data with a normal distribution was analyzed using one-way analysis of variance (ANOVA). Nonparametric data was analyzed using the Kruskal-Wallis test followed by Dunn’s post-test. Differences were statistically significant when *p*-value was less than 0.05.

## 3. Results

### 3.1. Peptide Concentrations in Plasma of Patients with CAD and Healthy Subjects

The basic clinical characteristics of the controls and patients with CAD are shown in Table 1. As can be seen from the table below, all four groups were well matched with respect to age, gender, smoking status, as well as BMI. It should be noted that patients younger than 50 years and older than 75 years old, with chronic inflammatory diseases, autoimmune diseases, acute infectious diseases, chronic liver, and kidney diseases, were not enrolled in this research. No significant differences in common biochemical characteristics were observed between the healthy subjects and patients either with stable or unstable angina pectoris. At the same time, patients with MI had a higher plasma concentration of total glycerides and a lower concentration of HDL cholesterol than did the controls and the other two CAD groups. The plasma concentration of total protein was also significantly elevated in the MI group, compared to both healthy subjects and patients with angina. The cardiac troponin I level, which is the biomarker of choice for the detection of cardiac injury, was notably increased in the plasma of patients with MI (Table 1).

The greatest peptide concentration was observed in the plasma of patients with MI. Thus, the plasma peptide level was almost 15 times higher in the MI group compared to the group of healthy subjects. We also found that plasma peptide concentrations were remarkably increased in patients with different angina pectoris states, compared to healthy volunteers. The plasma peptide level in patients with progressive unstable angina was six times higher than in control subjects, and almost 1.5 times higher than in patients with stable angina (Table 2).

### 3.2. The Levels of Some Soluble Anticoagulant As Well As Procoagulant Mediators Synthetized by Endothelial Cells Treated with Peptides Isolated from Plasma of Patients with CAD and Control Subjects

One of the major functions of endothelial cells is to produce physiologically important molecules, which have vasoprotective action, and are able to suppress platelet activation under physiological conditions [7,8,9]. Some molecules are expressed on the endothelial cell surface, while others are released in circulation. In this study, the levels of some soluble anticoagulant as well as procoagulant mediators synthetized by endothelial cells treated with peptides isolated from the plasma of patients with CAD and control subjects were examined. Since it has recently been shown that the HEPC culture can be considered as a robust and valid in vitro model suitable for studying endothelial cell function, we used the HEPC culture to evaluate the effects of peptide pools accumulated in the plasma of patients with CAD on endothelium. Postconfluent monolayers of HEPC were incubated with the peptide pools for 24 h. The possible cytolysis of endothelial cells in culture was measured by the release of cytosolic LDH into the culture supernatant. LDH release by the HEPC was not increased after 24 h exposure to peptides from both CAD patients and healthy volunteers.

#### 3.2.1. The Levels of Tissue Plasminogen Activator and Plasminogen Activator Inhibitor-1 Synthetized by Endothelial Cells Treated with Peptides Isolated from Plasma of Patients with CAD and Control Subjects

Endothelial cells are the major site of synthesis of tissue plasminogen activator (tPA), a 68-kDa glycoprotein, and a major mediator of endogenous fibrinolysis. The endothelium produces plasminogen activator inhibitor-1 (PAI-1), which serves to neutralize tPA activity [17,18]. Since these two compounds are involved in the maintenance of homeostatic balance and may play a role in cardiovascular disease development, we analyzed the levels of tPA and PAI-1 in endothelial cell cultures that were treated with peptides isolated from patients with CAD, and healthy volunteers. The results obtained are shown in Figure 1.

The tPA secretion level by endothelial cells exposed to treatment with peptides derived from the PUA group (after a 1 h incubation period) was slightly higher than that of the control group. When endothelial cells were treated with peptides derived from patients with MI, the tPA secretion level was slightly lower than in the control culture. The peptide pools isolated from healthy subjects and patients with SA did not cause changes of tPA secretion within 1 h. The stimulation of endothelial cells with peptides derived from the SA-group for 24 h was accompanied by a reduced amount of released tPA (Figure 1A). In addition, the level of secretion of PAI-1 was not influenced by the treatment of HEPCs with peptides isolated from both CAD patients and healthy volunteers (Figure 1B).

#### 3.2.2. The Levels of von Willebrand factor and Endothelin-1 Synthetized by Endothelial Cells Treated with Peptides Isolated from Plasma of Patients with CAD and Control Subjects

The von Willebrand factor (vWF) is one of the most important endothelium-derived molecules involved in controlling coagulation [19,20]. It plays a pivotal role in hemostatic plug formation, serving as a ligand for platelet adhesion to denuded vascular endothelium, and functions as a bridge molecule to support agonist-induced platelet aggregation. Thus, we explored the possible effect of peptides on vWF secretion by endothelial cells in vitro. The results obtained are shown in Figure 2A. The incubation of HEPCs with peptide pools isolated from either healthy subjects or patients with SA, revealed no effect on the vWF secretion level. However, when the HEPC culture was exposed to peptides derived from patients with PUA or MI, the secretion level of vWF was significantly increased within 24 h after stimulation (Figure 2A).

Endothelin-1 (ET-1) is one of the most potent endogenous vasoconstrictors secreted by endothelial cells, which acts as the natural counterpart of the vasodilator molecules. Apart from participating in the vascular tone regulation, ET-1 is involved in vascular remodeling, the development of inflammatory processes in the vascular wall, and contributes to both cell proliferation and apoptosis. Under physiological conditions, the effects of ET-1 are tightly regulated through the inhibition or stimulation of ET-1 release from endothelium [21]. Hemodynamic disorders are usually accompanied by imbalance in the production of vasodilator and vasoconstrictor agents. We examined the secretion of ET-1 by cultures of HEPCs treated with peptides isolated from healthy subjects and patients with CAD. The results obtained are shown in Figure 2B. The secretion of ET-1 decreased significantly within a 1 h treatment of endothelial cells with peptides derived from patients with CAD, compared to untreated culture (control). Thus, within 24 h of treatment with peptides derived from patients with SA, the level of ET-1 in the culture medium was in the same range as its level in unstimulated HEPC. However, treatment with peptides derived from patients with PUS caused a significant decrease in ET-1 level, and treatment with peptides derived from patients with MI was associated with a significant increase in ET-1 level (Figure 2B).

## 4. Discussion

There has been an increased interest in the study of peptides as possible effector molecules in many diseases and physiological conditions. Scientists have focused on the qualitative and quantitative analysis of the peptide spectrum in sera, or other biological fluids [22,23]. It has been established that peptides are synthesized in almost all organs and tissues of living organisms. The first group of peptides are intact small molecules (oxytocin, vasopressin, opioid peptides, ACTH, etc.,) that are released from larger precursor molecules during protein processing [24]. The second group of peptides represents “degradation products” that are formed as a result of the protein cleavage by the proteolytic enzymes [25].

The first group of peptides has been studied for decades, and it is now clear that their synthesis is usually highly regulated, since almost all of them have unique physiological functions and are involved in the maintenance of organism homeostasis, regulating processes such as proliferation, differentiation, cell death, and many others [22,26,27]. In contrast, the second peptide group has been considered to be a set of the degradation products of proteins, that have no specific function. However, recent experimental data have provided additional information on the proteolytic processes involved in the generation of components of “tissue-specific peptide pools” [14,28]. In the literature, several suggestions on ways of tissue specific peptide pool generation were described. Firstly, that peptides can be formed inside living cells by means of the cell specific proteinases, and further released into extracellular medium [29]. Secondly, that peptides can be formed extracellularly by means of the tissue specific proteolytic enzymes present in the extracellular matrix, such as matrix proteinases [30]. Finally, that peptides can be formed in dying cells and further released into surrounding fluids after cytolysis.

Recently, the development of peptide extraction methods and analytical technologies created an opportunity to identify numerous previously unknown peptides [31]. Despite the large repertoire of peptides that exist, the composition of tissue specific peptide pools is relatively stable under physiological conditions and does not show individual differences. The pathological processes and the use of medications affect the composition of peptide pools in target tissue. Therefore, peptides are convenient substances that can serve as biomarkers for early diagnosis in symptomatic patients, estimating disease progression, or monitoring responses to therapy.

During our experiments, we used platelet-depleted citrate plasma for the analysis of low molecular weight proteome [32]. The analysis of whole plasma is analytically challenging due to the wide range of protein and peptide concentrations. For example, albumin, the most abundant plasma protein, is present in the range of mg/mL, whereas peptides are more likely to be found in the pg/mL. Thus, it is important to remove high molecular weight proteins, such as albumin, immunoglobulins, transferrin, and lipoproteins, from plasma prior to the further analysis of plasma peptidome. Therefore, a unique technique described by Nikolaichyk for the analysis of peptide concentrations in the plasma of patients with CAD, and control subjects, has been employed [14].

We found that plasma peptide concentrations were remarkably increased in patients with cardiovascular disorders, in comparison with those in healthy donors (Table 2). In general, the elevated peptide levels in the plasma of CAD patients can be explained either by the alteration of protein metabolism, or by the intensification of processes associated with cell disruption. Many of the peptides formed under the pathological condition could be synthesized as byproducts of protein metabolism. Proteases, as well as specific cellular mechanisms such as protein ubiquitination, induce proteins to limited digestion generating intermediate peptides [33]. Cells can also produce peptides by directly translating small mRNA sequences [34]. The biological significance of peptides could be broad, modulating cell signaling from inside and outside the cells. Such untypical peptides can be formed specifically for some purpose, and act as modulating agents that could contribute to the maintenance of the homeostasis of living organisms, or may be recognized as risk factors involved in the progression of pathological processes.

The aim of this study was to investigate the effect of peptides derived from patients with CAD on HEPC. The study examined and compared the secretion levels of some molecules involved in the maintenance of vascular homeostasis by the cultures of human endothelial cells treated with peptides isolated from healthy subjects, and patients with CAD, namely, stable angina, progressive unstable angina, or myocardial infarction.

According to the results obtained, peptides isolated from healthy patients showed different effects on endothelial cell secretion compared to peptides isolated from patients with CAD. Moreover, peptides isolated from patients with SA, PUA, or MI, also had diverse effects (Figure 1 and Figure 2).

The most pronounced effects were observed under the incubation of HEPCs with peptides derived from patient with PUA or MI. Thus, peptides from patients with MI stimulated endothelial cells, resulting in the increased secretion of both vWF and endotelin-1. In contrast, the release of tPA by endothelial cells was inhibited by the presence of peptides derived from patients with MI in culture medium. The release of endothelin-1 may promote vasoconstriction, whereas the production of von Willebrand factor (vWF) shifts the hemostatic balance towards a procoagulant state, resulting in coronary arteries’ thrombosis, and subsequent ischemia. The treatment of endothelial cells with peptides derived from patients with PUA was accompanied by an increased secretion of tPA and vWF, while the amount of released ET-1 was decreased at the same time. The suppressive effect of peptides derived from patients with PUA on endotelin-1 production may be a part of the compensatory responses of organism to cardiovascular events.

## 5. Conclusions

Peptides from the plasma of patients with myocardial infarction promote endothelial cells to release both von Willebrand factor and endothelin-1; however, peptides from the plasma of patients with progressive unstable angina suppress the secretion of endothelin-1, and increase the secretion of von Willebrand factor and tissue plasminogen activator.

The molecular basis for these processes is not entirely clear, but it is reasonable to presume that the plasma peptides in patients with CAD may play a role in endothelial cell function.

We believe that the comparative analysis of the plasma peptides of patients with MI or angina, and healthy subjects, is a proper strategy for detecting either CAD biomarkers or triggers of cardiovascular events. Although several challenges will have to be met, the circulatory peptides are usually present in very low concentration, and there is high complexity involved due to the heterogeneity of CAD. Thus, there is a compelling need to develop sensitive, specific, and easily performed extracting and analytical techniques for plasma peptides analysis. Our immediate plans include using currently available approaches for cataloguing human blood plasma peptides, as well as detecting the individual variability, and the features of the peptides, related to cardiovascular diseases.

## Figures and Tables

**Figure 1 medicina-59-00238-f001:**
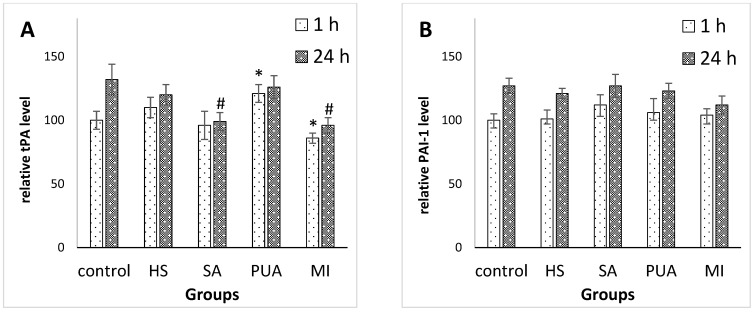
The relative levels of tPA (**A**) and PAI-1 (**B**) detected in HEPC culture incubated with peptide fraction isolated from plasma of healthy subjects (HS), patients with stable angina (SA), progressive unstable angina (PUA), or myocardial infarction (MI). The levels of both tPA and PAI-1 in culture treated with buffer instead of peptide solution (control) after 1 h incubation period were set at 100%, and all data shown represent percentage relative to control at 1 h time point. * *p* < 0.05 compared with control at time point 1 h; ^#^
*p* < 0.05 compared with control at time point 24 h.

**Figure 2 medicina-59-00238-f002:**
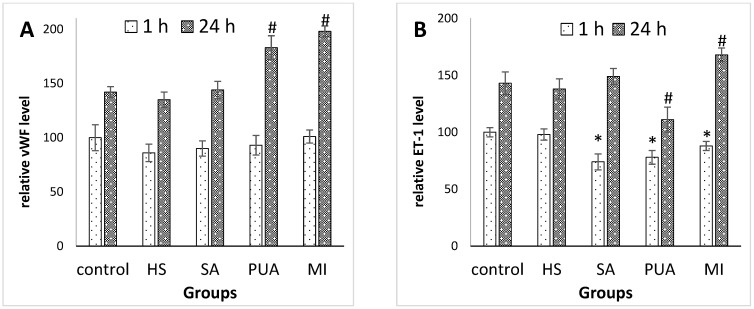
The relative levels of vWF (**A**) and ET-1 (**B**) detected in HEPC culture incubated with peptide fraction isolated from plasma of either healthy subjects (HS), patients with stable angina (SA), progressive unstable angina (PUA), or myocardial infarction (MI). The levels of both vWF and ET-1 in culture treated with buffer instead of peptide solution (control) after 1 h incubation period were set at 100%, and all data shown represent percentage relative to control at 1 h time point. * *p* < 0.05 compared with control at time point 1 h; ^#^
*p* < 0.05 compared with control at time point 24 h.

**Table 1 medicina-59-00238-t001:** The main clinical characteristics of patients with CAD, and healthy subjects.

	Healthy Subjects (*n* = 20)	Patients with Stable Angina (*n* = 25)	Patients with Progressive Unstable Angina (*n* = 28)	Patients with Myocardial Infarction (*n* = 17)
Age, years	53 ± 6	63 ± 8	67 ± 7	61 ± 7
Sex, male, *n* (%)	10 (50)	15 (60)	14 (50)	11 (65)
Smoking, *n* (%)	7 (35)	9 (36)	11 (39)	6 (35)
BMI, kg/m^2^	27 ± 4	29 ± 6	29 ± 3	28 ± 6
TC, mmol·L^−1^	5.14 ± 0.99	5.96 ± 1.64	6.39 ± 2.18	5.28 ± 0.94
TG, mmol·L^−1^	1.28 ± 0.43	2.50 ± 0.70	2.21 ± 1.12	3.29 ± 1.17 *
LDL, mmol·L^−1^	2.38 ± 0.47	3.39 ± 1.38	3.75 ± 1.19	3.29 ± 1.17
HDL, mmol·L^−1^	1.64 ± 0.45	1.64 ± 0.87	1.31 ± 0.51	0.85 ± 0.24 *
TP, g·L^−1^	13.8 ± 4.0	10.4 ± 5.0 ^§^	11.5 ± 4.6 ^§^	35.5 ± 11.8 *
hs-cTnI, pg·mL^−1^	≤5	≤5	17.6 ± 9.2 *^#§^	24.3 ± 10.2 *^#^

Values are expressed as mean ± SD; * *p* < 0.05 significantly different from control subjects; ^#^
*p* < 0.05 significantly different from group of patients with stable angina; ^§^
*p* < 0.05 significantly different from group of patients with myocardial infarction. TC = total cholesterol, TG = total glycerides, LDL = low density lipoproteins, HDL = high density lipoproteins, TP = total protein, hs-cTnI = high sensitive cardiac troponin I.

**Table 2 medicina-59-00238-t002:** The concentration of peptides in plasma of patients with CAD, and healthy subjects.

Groups	Peptide Concentration, μg per mL of Plasma
Healthy subjects (*n* = 20)	68 ± 18
Patients with stable angina (*n* = 25)	336 ± 17 *^§^
Patients with progressive unstable angina (*n* = 28)	428 ± 27 *^#§^
Patients with myocardial infarction (*n* = 17)	987 ± 64 *^#^

Values are expressed as mean ± SD; * *p* < 0.05 significantly different from control subjects; ^#^
*p* < 0.05 significantly different from group of patients with stable angina; ^§^
*p* < 0.05 significantly different from group of patients with myocardial infarction.

## Data Availability

Data supporting the reported results are available on request.

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
