# Peer review of "Effect of Peptides from Plasma of Patients with Coronary Artery Disease on the Vascular Endothelial Cells"

_medicina, 2023, doi:10.3390/medicina59020238_

Round 1
Reviewer 1 Report
This study by Kozyk and colleagues assesses plasma peptide concentrations in patients with either stable angina or acute coronary syndromes and a control group, comparing peptide values between the control group and CAD patients in stable or acute phase. Furthermore, the effect of peptides on human endothelial progenitor cells has been assessed through the measurement of tPA, PAI-1, vWF and ET-1. Although the authors quite clearly describe their methods and results, there are some issues that should be addressed.
Major points:
1. The authors state in their Introduction, page 1, lines 38-41: “It is not clear exactly how atherosclerosis starts or what causes it, though it is well known that coronary endothelial dysfunction contributes to both initiation and progression of atherosclerotic plaques”. This statement should be corrected, as many CVD risk factors have been well described as well as their contribution to CVD pathophysiology.
2. The timepoint when plasma samples were obtained is not provided in the Methods section and should be mentioned.
3. The authors describe that the procedures followed for plasma peptide isolation are based on the methods of Nikolaichyk et al (related to reference 12). However, I could not find neither in the references listed nor in the current literature any relevant article.
4. Kozyk et al. compare peptide concentrations between ACS groups and the control group as also between two ACS groups. MI patients seem to have significantly higher peptide concentration than PUA, SA patients and controls. However, no table including basic characteristics per patient group such as age, gender, prevalence of traditional CVD risk factors is provided. In the Methods section, the control group is referred to be matched for these factors with the ACS patient groups. Are the ACS patient groups matched for these factors between them?
5. As mentioned in the Methods section page 3, lines 103-104: “Peptide solutions from five individuals of the same experimental group were randomly selected and used in further analysis”. This should be noted in the Discussion section as a limitation of the study, as the small sample size (n=5 for each group) may influence the results of the subsequent analysis. If possible, the authors are encouraged to proceed with assessing the effects of peptide pool on HEPCs using peptide pools from all the participants.
6. In the section 3.2 all the comparisons presented are correctly made between patient and control groups. We observe an unexpected increase in relative tPA, PAI-1, vWF and ET-1 levels between the 1h and 24h timepoints in the control group. Is this increase significant? What could lead to this increase? Could this be attributed to the small sample size (n=5 for each group), also raised in my previous comment?
7. The Discussion section should be improved. Interesting results presented in section 3.1 are not adequately discussed. MI patients have significantly higher peptide concentrations (more than double compared to PUA patients, almost triple compared to SA patients and about 15-fold compared to controls). Could this be attributed only to peptides released because of myocardial necrosis? The authors should also elaborate on the novelty of their study. Few studies assessing plasma peptidome have been published concerning CVD.
Minor points
1. An image summarizing the work done by the authors would be appreciated and could help the reader better understand the article in a concise way.
2. Please correct some typos/ grammar errors
· page 2, line 1 A research
· page 3, line 139 anti-endotelin-1
· page 7, line 272 data has
· page 7, line 300 patients with CADs
· page 7, line 309 of tAP etc.
Author Response
Dear Editor,
We carefully reviewed each of the comments made by the reviewers and made edits to the manuscript. The comments and suggestions made by the reviewers addressed key aspects of our study, we feel that our manuscript is much improved now and we thank the reviewers for their helpful comments. We have addressed each point and made changes to our manuscript in yellow. The reviewer’s comments are addressed below.
From the authors,
Dr. Tetiana Halenova, Dr. Marko Kozyk, Dr. Kateryna Strubchevska

Reviewer 2 Report
In materials and methods:
Chapter 2.1.
Describe in which tubes the blood was taken and how the plasma was obtained.
Chapter 2.2.
For each chemical, the producer and the country of the producer of the chemical should be indicated as described in chapter 2.3.,
Chapter 2.4.
Why were not commercially available ELISA tests used for the mentioned parameters, but in-house tests?
Chapter 2.5.
Explain why the results are presented with means and standard deviations and not median and interquartile range if the number of respondents per group is less than 30? Given that the number of subjects is <30, results should be presented as median and IQR, and statistical tests should be non-parametric. Is the p-value corrected for multiple testing? If not, you should.
In the results:
How was the concentration of the peptide from table 1 determined?
In the results, only the results are described, they are not discussed. Please rewrite the results chapter.
Concentrations of selected markers are relativized according to what? Please explain in more detail.
In the discussion:
Please speculate in more detail which peptides could have an effect on the vascular endothelium.
Author Response
The authors would like to thank the reviewer for their precious time and invaluable comments. We have carefully addressed all the comments. The responses and explanations related to your comments are listed below. The corresponding changes and refinements made in the revised paper (highlighted in yellow).

Round 2
Reviewer 1 Report
The manuscript has been significantly improved. There are additional important
revisions required though, based on my original comments that were not all adequately addressed.
1) You have correctly added Table 1. However, no information concerning smoking and DM status is provided. Please modify the table 1 accordingly and comment accordingly.
2) I understand that the deadline may have been strict. However, I consider adding a relevant image necessary for your article. Please add it as requested.
3) The discussion section significantly improved. However, I consider important that you further comment on the novelty of your article as well as possible limitations of the study.
4) Please note in the Methods section the kit used for troponin measurements and the 99th percentile value for the relevant kit, beside reference range.
5) Please correct some typos/ grammar errors
Line 102 Katrii et al.
Line 330 in comparison with these (instead of with that)
Line 339 could contribute (instead of could be contribute)
Author Response
Dear Reviewer,
We corrected everything according to your comments.
Thank you!
Authors team
Reviewer 2 Report
Dear Authors,
thank you for your answers and corrections.
Congratulations for your great work. Continue in this way.
Author Response
Dear Reviewer,
Thank you very much!
Yours sincerely,
Authors team